# A Facile Fabrication of Ordered Mesoporous Carbons Derived from Phenolic Resin and Mesophase Pitch via a Self-Assembly Method

**DOI:** 10.3390/nano12152686

**Published:** 2022-08-04

**Authors:** Jae-Yeon Yang, Tae Hoon Ko, Yun-Su Kuk, Min-Kang Seo, Byoung-Suhk Kim

**Affiliations:** 1Convergence Research Division, Korea Carbon Industry Promotion Agency (KCARBON), 110-11 Banryong-ro, Deokjin-gu, Jeonju-si 54853, Jeollabuk-do, Korea; 2Department of Nano Convergence Engineering, Jeonbuk National University, 567 Baekje-daero, Deokjin-gu, Jeonju-si 54896, Jeollabuk-do, Korea; 3Department of Organic Materials & Fiber Engineering, Jeonbuk National University, 567 Baekje-daero, Deokjin-gu, Jeonju-si 54896, Jeollabuk-do, Korea

**Keywords:** mesoporous carbons, self-assembly method, mesophase pitch, carbon electrode

## Abstract

Ordered and disordered mesoporous structures were synthesized by a self-assembly method using a mixture of phenolic resin and petroleum-based mesophase pitch as the starting materials, amphiphilic triblock copolymer F127 as a soft template, hydrochloric acid as a catalyst, and distilled water as a solvent. Then, mesoporous carbons were obtained via autoclave method at low temperature (60 °C) and then carbonization at a relatively low temperature (600 °C), respectively. X-ray diffraction (XRD), small-angle X-ray scattering (SAXS), and transmission electron microscopy (TEM) analyses revealed that the porous carbons with a mesophase pitch content of approximately 10 wt% showed a highly ordered hexagonal mesostructure with a highly uniform pore size of ca. 5.0 nm. In addition, the mesoporous carbons prepared by self-assembly and low-temperature autoclave methods exhibited the amorphous or crystalline carbon structures with higher specific surface area (SSA) of 756 m^2^/s and pore volume of 0.63 cm^3^/g, depending on the synthesis method. As a result, mesoporous carbons having a high SSA were successfully prepared by changing the mixing ratio of mesophase pitch and phenolic resin. The electrochemical properties of as-obtained mesoporous carbon materials were investigated. Further, the OMC-meso-10 electrode delivered the maximum SC of about 241 F/g at an applied current density of 1 A/g, which was higher than those of the MC-10 (~104 F/g) and OMC-20 (~115 F/g).

## 1. Introduction

An electric double-layer capacitor (EDLC) or a supercapacitor (SC) as a sustainable and renewable energy resource has given significant focus due to its excellent power density, superior cycle life, and rapid discharge/charge rate, etc., which can supplement secondary batteries [1,2,3,4]. Activated carbon (AC) is mainly used as an active material for EDLCs fabrication due to its large specific surface area and low cost. However, AC has a low electrical conductivity of 0.1–1 S cm^−1^, which increases the internal resistance of the EDLCs. Moreover, AC mainly contains micropore (<2 nm), which restricts access to electrolytes, because of large ionic sizes of organic electrolytes [5,6].

Ordered mesoporous carbon (OMC) materials show many attractive properties including good conductivity, electrochemical properties, tunable pore structure, and high specific surface area (up to 1100 m^2^ g^−1^), and can be therefore considered as a favorable electrode material for EDLCs [7,8,9,10]. OMC has been extensively studied for its potential usages in sorption, catalysis, supercapacitors, and electrocatalysts. To date, OMCs with different mesoporous structures have been effectively synthesized by both hard and soft template methods. However, the main drawbacks of the hard-template method are that it requires multiple steps for scaffold preparation and needs to use sacrificial surfactant templates as scaffolds. The high cost and complex synthesis of OMCs significantly have brought restrictions on application. In contrast, the soft-template method, in which amphiphilic surfactant compounds such as block copolymers are used as soft templates based on the principle of liquid-crystal templating, can produce OMC in a more flexible manner [11,12,13]. Recently, mesophase pitches have been applied as carbon sources to prepare mesoporous carbon (MC) materials. Compared with other carbon precursors, mesophase pitches can produce high-quality graphitized carbon. Generally, the pitch powders were mixed with a soft template and phenol resin at a slightly higher temperature than the softening points. In this case, the pitch particles were self-assembled into the template to form mesoporous composites. Finally, the mesoporous carbon was derived after the elimination of the template [14,15,16,17]. 

In this study, OMC and disordered MC were obtained by an organic–organic self-assembly process using a mixture of phenol resin and petroleum-based mesophase pitch as the carbon precursors at a low-temperature autoclave (LTA), and subsequently carbonized. The crystalline structures, textural properties, and electrochemical performances were investigated. 

## 2. Materials and Methods

### 2.1. Preparation of Mesoporous Carbons

OMCs were synthesized by LTA method. First, 3.20 g of phenolic resin powder (CB-8081, Kangnam Chemical. Co., Ltd., Seoul, Korea) and 4.8 g of F127 (poly(ethylene glycol)-*block*-poly(propylene glycol)-*block*-poly(ethylene glycol) diacrylate, Sigma-Aldrich, St. Louis, MO, USA) were added in mixed solvent (80 mL, 1:1 water/ethanol, *v*/*v*) and stirred for 60 min to obtain clear solution. Then, 0.387 g of 37% HCl solution and different amounts (0, 10, 20 wt%) of mesophase pitch [17] were added to the above solution. The physical properties and elemental composition of the mesophase pitch used in this work are presented in Table 1. After stirring for 1 h, 37% formaldehyde (4.8 g, Sigma-Aldrich, St. Louis, MO, USA) solution was injected slowly under stirring. It continued to be stirred until a yellowish homogeneous solution was obtained. Then, it was poured into a Teflon-lined autoclave reactor and kept in electric oven for 3 days at 60 °C. The obtained polymeric monolith was collected by filtration, washed with deionized water, dehydrated in an oven at 60 °C, and further retained at 80 °C for 12 h, which led to a color change of the polymeric monolith into salmon pink. Subsequently, the products were carbonized in a tubular furnace under an inert atmosphere (N_2_ flow) by heating the products at a heating rate of 1 °C/min up to 600 °C and then maintained further for 6 h. Here, it should be noted that soft template was easily removed by conventional calcination at 600 °C [18,19]. The final products were denoted as OMC-meso-x (here, *x* denotes the amount of mesophase pitch used. *x* = 0, 10, 20 wt%). The type of precursors and synthesis conditions are presented in Table 2. 

For comparison, MC samples were also prepared by an evaporation-induced self-assembly (EISA) method, except that the uniform yellowish solution was poured into the molds of various shapes and further retained at ambient temperature for 3 days to evaporate the ethanol. The other conditions of synthesis were the same as above. The obtained monolith showed a dark red color and was carbonized at 600 °C for 6 h at a heating rate of 1 °C/min. The carbonized products were designated as MC-meso-*x*, as above. The schematic of the synthesis preparation was shown in Figure 1. 

### 2.2. Characterization

Powder X-ray diffraction (XRD) analysis was performed under a PANalytical NLD diffractometer at a scanning rate of 2°/min. Small-angle X-ray scattering (SAXS) studies were carried out on an Anton Paar SAXSpace small-angle X-ray scattering equipment. Raman spectra were recorded with a Raman spectrometer (FEX, Nost, Korea) using a 531 nm line as the excitation source at room temperature. The surface structure of the MCs was characterized by field-emission scanning electron microscopy (FE-SEM, SU8820, HITACHI, Japan). Transmission electron microscopy (TEM) was performed on a Jeol JEM F2010 microscope. The specimens for TEM analysis were performed by ultrasonically diluting the samples in isopropanol and drop-casting on copper grids covered with a carbon film. Nitrogen adsorption isotherms were recorded at 77 K using a Micromeritics ASAP 2020 device (USA). Prior to nitrogen adsorption, the products were vacuum-dried at 300 °C for 6 h. The specific surface areas of the carbon products were derived by the Brunauer–Emmett–Teller (BET) technique. 

The electrochemical studies, including cyclic voltammograms (CVs) and galvanostatic charge-discharge (GCD) tests, were performed using an electrochemical workstation (Versastat 4) at room temperature under a three-electrode configuration. The MC or OMC modified electrodes, Hg/HgO (1 M NaOH), and platinum foil (1.5 × 1.5 cm) were used as working, reference, and counter electrodes, respectively. The CV curves were recorded in the potential window between −1.0 to 0 V in 6M KOH at different scan rates of 5–200 mV s^−1^. The charge–discharge properties of the MC and OMC materials were verified by the GCD test. The specific capacitance (*SC*, F g^−1^) was obtained using the following equation, [20,21]: (1)SC=(I×Δt)/(m×ΔV)
where *I* is the applied current density for GCD (A), Δ*t* is the required time for discharge (s), *m* is the active mass (g) of the electrode material (MC and OMC), and Δ*V* is the active potential window (V) in GCD profile. The electrochemical impedance measurements were recorded in the frequency range between 100 kHz to 1 Hz at open-circuit voltage and the data were presented in Nyquist plots. The EIS graphs were fitted with Randles equivalent electrical circuit using Zsimpwin software (VersaStudio, PowerSINE, Princeton Applied Research Co., Oak Ridge, TN, USA). 

## 3. Results and Discussion

### Structure and Morphology

Figure 2a displays the small-angle XRD patterns of the MCs and OMCs prepared with different amounts of mesophase pitch. The SAXS spectra of MC-meso-10 and OMC-meso-10 exhibited the typical low-angle peaks at 2θ = 0.50° and 0.75°, respectively, indicating an ordered mesoporous structure [22,23]. In particular, the OMC-meso-10 showed sharp intensity, implying a more ordered mesopore structure, whereas the OMC-meso-20 showed rather broad peak due to the weak long-range ordering of its mesopore structure [22,23,24]. Meanwhile, the wide-angle XRD patterns of the MCs and OMCs (Figure 2b) exhibited broad diffraction peaks at 2θ = 23.5° and 43.2°, corresponding to the (002) and (101) reflections of amorphous carbon, respectively, suggesting that the carbons prepared at a low carbonization temperature of 600 °C had lower crystallinity and thus lower graphitization [24,25,26,27]. Raman spectra were widely used for studying carbon nanostructures. Raman spectra determined the degree of graphitization roughly from the *I_D_/I_G_* value. Here, the band at around 1580 cm^−1^ was assigned to the G (graphite) band that originates from the stretching mode of sp^2^ hybridized orbitals of carbon–carbon bonds. The band at around 1360 cm^−1^ was assigned to the D (disorder) band, which originated from the breathing mode of sp^2^ hybridized carbons. Figure 2c presents the Raman spectra of the MCs and OMCs, respectively. Overall, the mesophase pitch increased, the values of *I_D_/I_G_* decreased from 1.05 to 0.91 for the MCs and from 1.04 to 0.84 for the OMCs, respectively, indicating an increased ordered mesostructure, although the OMC-meso-20 showed slightly increased *I_D_/I_G_* value (~0.89), compared to the OMC-meso-10 (~0.84), probably due to the interruption by the excess amount (~20 wt%) of mesophase pitch [28,29,30]. This result was further confirmed by the FE-SEM (Figure 3) and TEM (Figure 4) analysis. Figure 3 depicts the FE-SEM images of the MCs and OMCs prepared with different amounts of mesophase pitch. Obviously, the OMC-meso-10 exhibited a typical stripe-like and hexagonally arranged mesopore, which was well consistent with the high-quality hexagonal meso-structure [31,32]. It was further confirmed by TEM analysis. As shown in Figure 4, the OMC-meso-10 showed an ordered hexagonal arrangement of mesopores (particularly, a long-range hexagonal arrangement) with the (001) and (110) directions, which indicates a highly ordered mesostructure [20]. Interestingly, the OMC-meso-20 exhibited worm-like structures with less mesopore patterning. The result suggested that the incorporation of the mesophase pitch could control the ordering of the mesoporous structure and microstructure of the MCs and OMCs [33,34,35,36]. 

Further, the pore structure of the obtained mesoporous carbon materials was analyzed by nitrogen sorption method. Figure 5 displays the nitrogen adsorption–desorption isotherms (a) of as-obtained materials and the resultant pore-size graphs (b). The detailed textural characteristics are given in Table 3. The OMCs and MCs exhibited characteristic type-IV adsorption isotherms with clear hysteresis loops (Figure 5a), which indicates the presence of well-organized mesoporous patterns with cylindrical canals [37,38,39]. The pore-size distributions of the OMCs and MCs were calculated from the adsorption branches by the Barrett–Joyner–Halenda method. As shown in Figure 5b, the OMCs and MCs exhibited a narrow peak with a uniform mesopore size from 5.00 nm to 6.44 nm. Further, the OMC and MC showed larger BET surface areas of 500 and 756 m^2^/g and total pore volumes of 0.33 and 0.65 cm^3^/g, respectively [40,41].

In order to further verify the energy-storage properties of as-prepared mesoporous carbons (MCs, OMCs), CV and GCD tests were carried out. The MC-modified electrodes were initially accessed by the CV method. The CV curves were recorded in the potential window between −1.0 to 0 V at different scan rates (5–200 mV s^−1^) in 6 M KOH. Figure 6 shows the CV responses of MC-meso-10 (Figure 6a), OMC-meso-10 (Figure 6b) and OMC-meso-20 (Figure 6c), respectively. The pseudo-rectangular behavior of CV curves was observed for all the MC-modified electrodes at different scan rates, confirming the electric double-layer capacitive (EDLC) nature of as-prepared carbon materials [42,43]. That is, the charge storage occurs via a double-layer formation between the interface of the electrode and electrolytes without any faradic process. Further, the similar pseudo-rectangular CV characteristics were observed even at a high scan rate (200 mV/s) for all the MC-modified electrodes, suggesting the good rate capability of the carbon materials for high-performance supercapacitor applications. Figure 6d shows the comparative CV behavior of MC-meso-10-, OMC-meso-10- and OMC-meso-20-modified electrodes recorded at a scan rate of 100 mV/s. The comparative CV curves clearly confirmed that the OMC-meso-10-modified electrode exhibited slightly higher CV integral area when compared to that of MC-meso-10 and OMC-meso-20. These preliminary CV studies clearly demonstrated that as-prepared carbon materials were potentially useful for high-performance supercapacitor applications [44,45,46]. In order to validate the real supercapacitor performance of as-prepared porous carbon materials, GCD studies were carried out. Figure 7 shows the GCD curves of MC-meso-10 (Figure 7a), OMC-meso-10 (Figure 7b), and OMC-meso-20 (Figure 7c) at different applied current densities from 1 to 10 A/g in 6 M KOH. The GCD curves nearly exhibited standard ∧-shaped behavior for all the carbon materials, implying that the charge storage mainly occurred through EDLC process [46,47,48]. The comparative GCD curves for all the carbon materials at an applied current density of 1 A/g are displayed in Figure 7d. It was clearly observed that the OMC-meso-10-modified electrode exhibited the longer discharge time when compared to MC-meso-10- and OMC-meso-20-modified electrodes, which was also well-matched with CV results. The calculated SC values vs. current density for all the carbon materials are plotted in Figure 8a. As expected, the OMC-meso-10-modified electrode delivered a high SC value for all the applied current densities when compared to other carbon-modified electrodes. The maximum SC of OMC-meso-10 electrode was about 241 F/g at an applied current density of 1 A/g, which was more than 2 times higher than those of the MC-10 (~104 F/g) and OMC-20 (~115 F/g). The higher SC value of OMC-meso-10 was due to the high micropore volume ratio (~42.42%) and the presence of a highly ordered mesoporous structure with a high surface area, which enhances the accumulation of electrolytes at the electrode interface in large quantities and also promotes efficient ionic transfers between the electrode and electrolyte [9,11]. These electrochemical studies clearly demonstrated that the prepared mesoporous carbon materials were highly useful for real energy-storage applications. 

Further, the electrochemical properties of the mesoporous carbon-modified electrodes were studied by electrochemical impedance measurements. Figure 8b shows the Nyquist plot of different mesoporous carbon-modified electrodes. It presented the semicircle at a higher-frequency region, with the linear response at lower-frequency regions. The diameter of the semicircle was directly related to the insulating characteristics of the materials [49,50]. The as-obtained Nyquist curve fitted with Randles equivalent electrical circuit was used to evaluate the solution resistance (R_S_) and charge transfer resistance (R_CT_) of the mesoporous carbon-modified electrodes using Zsimpwin software (VersaStudio, PowerSINE, Princeton Applied Research Co., USA). The OMC-meso-10 showed the low R_S_ and R_CT_ values (0.44 Ω, 0.06 Ω), when compared to MC-meso-10 (0.62 Ω, 1.08 Ω) and OMC-meso-20 (0.54 Ω, 0.36 Ω) electrodes. These lower R_S_ and R_CT_ values of OMC-meso-10 were further evidenced for obtaining higher electrochemical energy-storage performance. 

## 4. Conclusions

In this study, MCs and OMCs with high SSA were prepared by a self-assembly process and a low-temperature autoclave (LTA) method using the mixture derived of phenolic resin and different amounts of petroleum based-mesophase pitch as the carbon precursors, respectively. Small-angle XRD analysis revealed that the OMC prepared with 10 wt% mesophase pitch (OMC-meso-10 electrode) had a highly ordered hexagonal meso-structure. This was further confirmed by TEM analysis, which revealed the presence of stripe-like and hexagonally arranged mesopores, indicating the formation of a high-quality hexagonal mesostructure. In addition, the mesoporous carbons prepared by LTA methods exhibited the crystalline carbon structures with higher specific surface area (SSA) of 756 m^2^/s and pore volume of 0.63 cm^3^/g. Further, the OMC-meso-10 electrode delivered the maximum SC of about 241 F/g at an applied current density of 1 A/g, which was higher than those of the MC-10 (~104 F/g) and OMC-20 (~115 F/g). As a result, the LTA method could offer useful advantages over the EISA method in the construction of monolithic carbon products with an ordered mesostructure, and also might be an efficient technique for the preparation of the mesoporous carbons with high SSA in industrial applications. 

## Figures and Tables

**Figure 1 nanomaterials-12-02686-f001:**
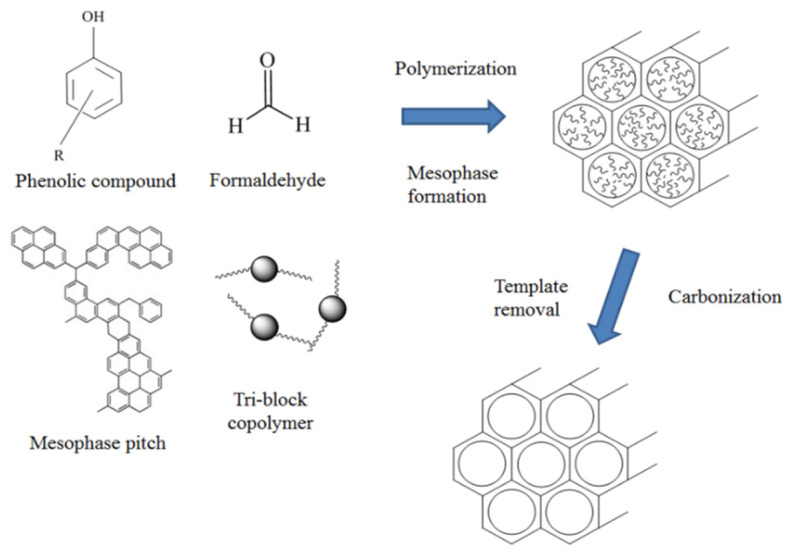
Schematic of the formation mechanism of self-assembled mesoporous carbons.

**Figure 2 nanomaterials-12-02686-f002:**
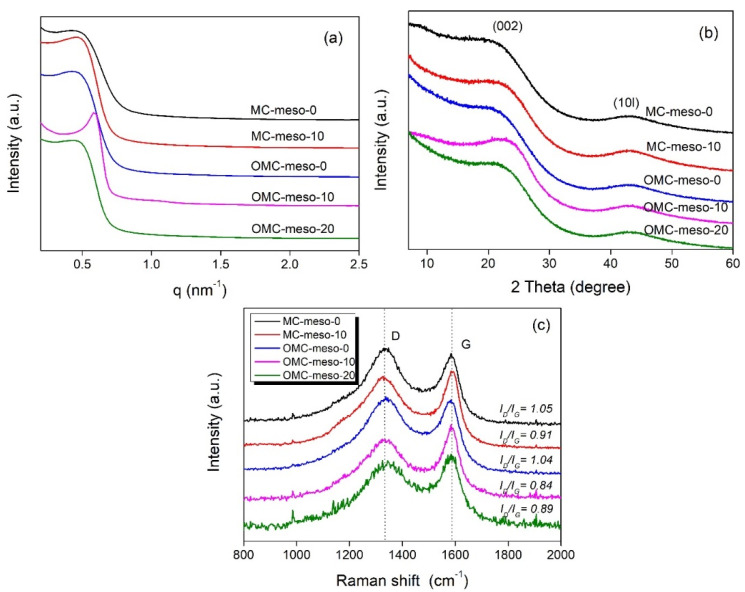
Small-angle (**a**) and wide-angle (**b**) XRD patterns and Raman spectra (**c**) of MCs and OMCs prepared with different amounts of mesophase pitch.

**Figure 3 nanomaterials-12-02686-f003:**
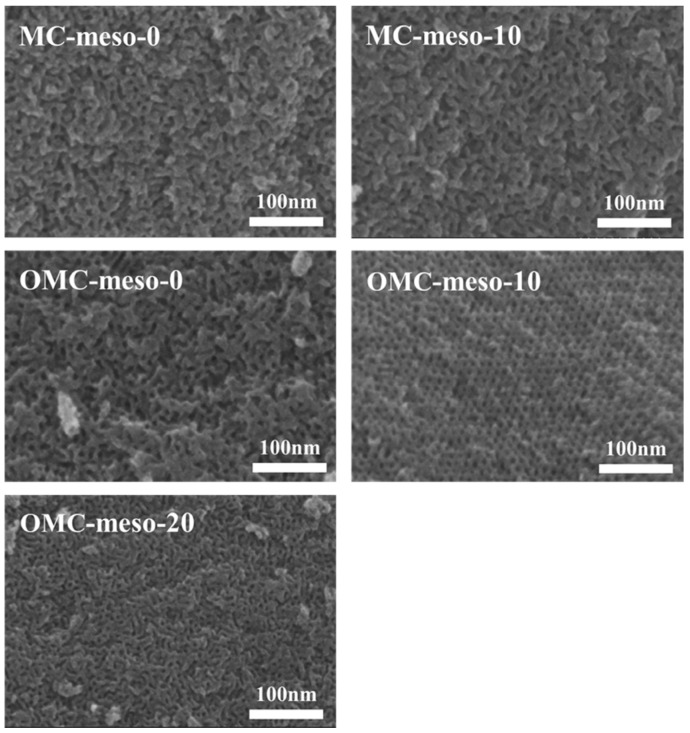
FE-SEM images of MCs and OMCs prepared with different amounts of mesophase pitch.

**Figure 4 nanomaterials-12-02686-f004:**
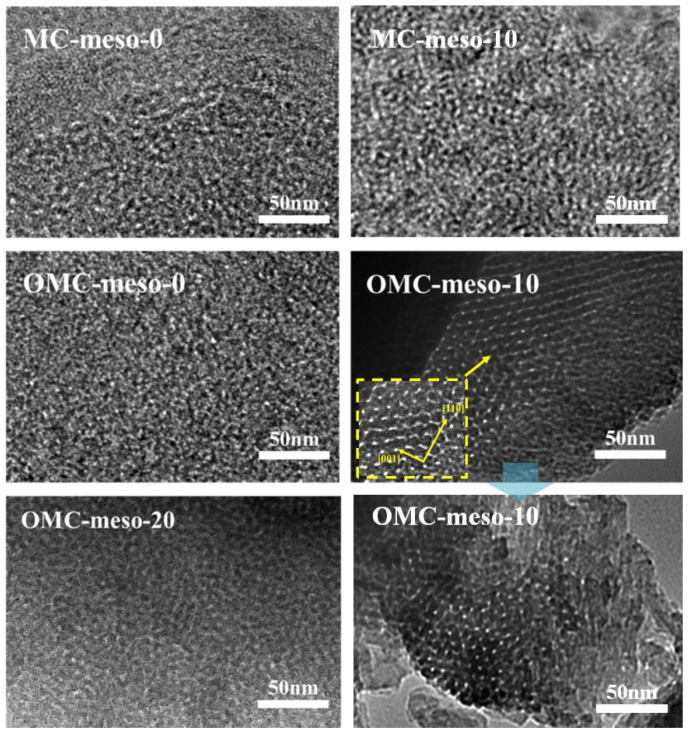
TEM images of MCs and OMCs prepared with different amounts of mesophase pitch.

**Figure 5 nanomaterials-12-02686-f005:**
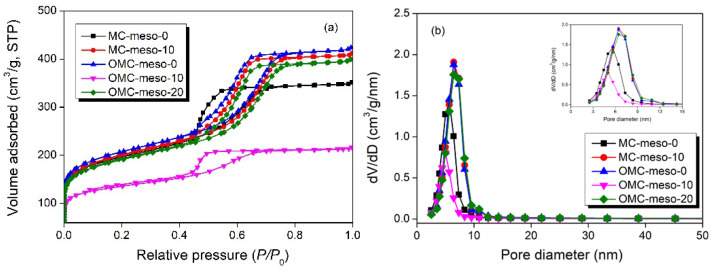
Nitrogen sorption isotherms (**a**) and pore-size distribution curves (**b**) of MCs and OMCs prepared with different amounts of mesophase pitch.

**Figure 6 nanomaterials-12-02686-f006:**
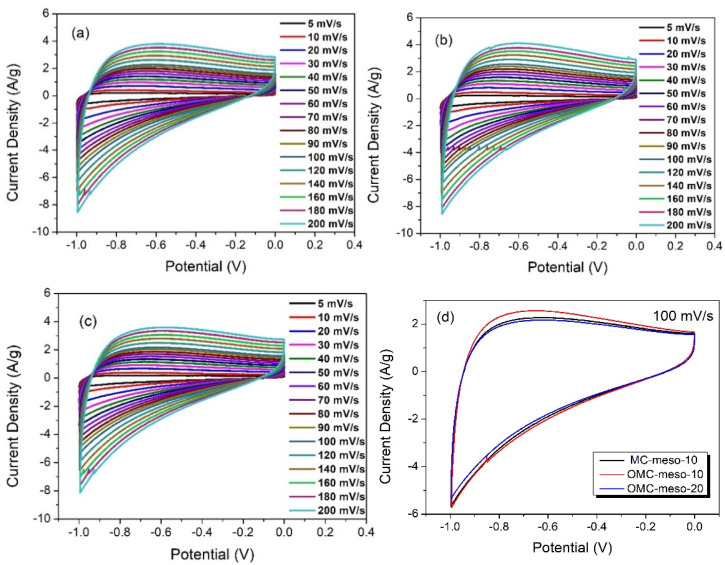
CV curves of MC-meso-10 (**a**), OMC-meso-10 (**b**), OMC-meso-20 (**c**) at scan rates from 5 to 200 mV/s, and comparative CV curves (**d**) of MC and OMCs at 100 mV/s.

**Figure 7 nanomaterials-12-02686-f007:**
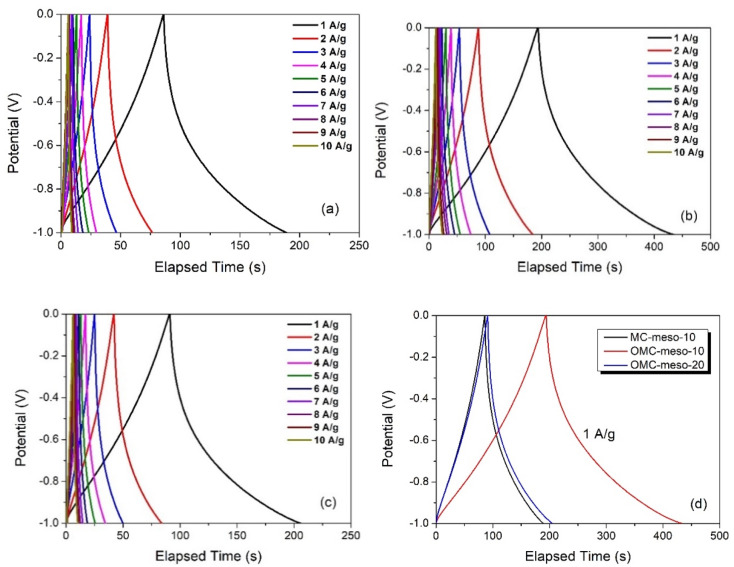
GCD curves of MC-meso-10 (**a**), OMC-meso-10 (**b**), OMC-meso-20 (**c**) at different current densities from 1 to 10 A/g, and comparative GCD curves (**d**) of MC and OMCs at current densities of 1 A/g.

**Figure 8 nanomaterials-12-02686-f008:**
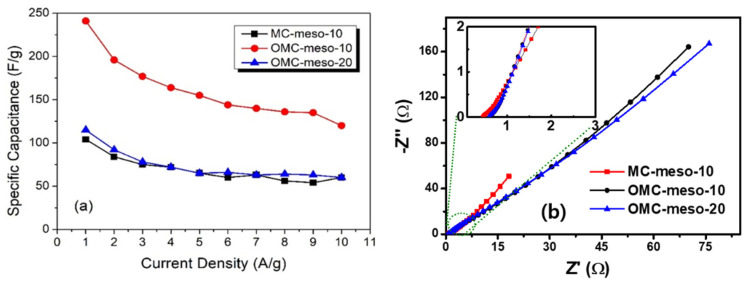
Specific capacitances at different current densities (**a**) and Nyquist plots (**b**) of MCs and OMCs prepared with different amounts of mesophase pitch.

**Table 1 nanomaterials-12-02686-t001:** Physical properties and elemental composition of mesophase pitch.

Samples	Softening Point (°C)	Carbon Yield (%)	Elemental Composition (%)				C/H Ratio	Quinoline Insoluble (%)
			C	H	N	S		
Mesophase pitch	227	67.66	94.78	4.752	0.105	0.279	1.67	35.93

**Table 2 nanomaterials-12-02686-t002:** Types of precursors and synthesis conditions of MCs and OMCs ^1^.

Samples	Carbon Source	Method
MC-meso-0	Phenolic resin	Self-assembly
MC-meso-10	Phenolic resin + Mesophase pitch 10 wt%	Self-assembly
OMC-meso-0	Phenolic resin	Hydrothermal
OMC-meso-10	Phenolic resin + Mesophase pitch 10 wt%	Hydrothermal
OMC-meso-20	Phenolic resin + Mesophase pitch 20 wt%	Hydrothermal

^1^ Structure-directing agent: F127; solvent: ethanol+H_2_O; synthesis temperature: 60 °C; synthesis time: 3 days; carbonization temperature: 600 °C.

**Table 3 nanomaterials-12-02686-t003:** Textural and structural properties of MCs and OMCs prepared with different amounts of mesophase pitch.

Samples	S_BET_ (m^2^/g)	V_total_ (cm^3^/g)	V_micro_ (cm^3^/g)	V_meso_ (cm^3^/g)	D_BJH_ (nm)
MC-meso-0	711	0.54	0.18	0.36	5.66
MC-meso-10	729	0.63	0.18	0.45	6.44
OMC-meso-0	756	0.65	0.18	0.47	6.34
OMC-meso-10	500	0.33	0.14	0.19	5.00
OMC-meso-20	696	0.62	0.18	0.44	6.44

## Data Availability

All data are available upon reasonable request.

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
