# Peer review of "A Facile Fabrication of Ordered Mesoporous Carbons Derived from Phenolic Resin and Mesophase Pitch via a Self-Assembly Method"

_nanomaterials, 2022, doi:10.3390/nano12152686_

Round 1

Reviewer 1 Report

This manuscript demonstrates the successful preparation of OMCs with high specific surface area using a mixture of phenolic resin and mesophase pitch as carbon precursors. The topic is closely associated with the scope of Nanomaterials but there are still some details needs to be modified before it could be considered for publication. The comments are given as follows.

Major points

1.     The ordered mesoporous formation mechanism and removal method of soft template are not clear in the present paper. Please explain it further.

2.   The perspective of OMC-meso-10 in Figure 4 is not consistent with other images in Figure 4, and the magnification of MC-meso-10 images are also inconsistent with other images in Figure 4. It is strongly suggested to make them consistent for better comparison.

3.   Raman spectrum should be carried out to further prove the crystallinity and amorphization of mesoporous carbon in Figure 2.

4.   The frequency range of electrochemical impedance spectrum tests should be provided to facilitate the analysis of the high frequency region of Nyquist curve in Figure 8.

Minor points

1.   The MCs at the beginning of Part 2.1 suggest to be revised to OMCs.

2.   All the fuzzy insets in Figure 5 should be replaced.

3.   The language in the main text and the image formats need to be improved. Especially, the scan rates in Figure 6 and the current densities in Figure 7 are not consistent with their Figure Captions. Please revise carefully.

Reviewer 2 Report

This project reports the development of ordered mesoporous carbon using mesophase pitch and phenolic resin as the carbon precursors and the block copolymer F127 to create the mesopores. The prepared ordered mesoporous carbon has been characterized by using various techniques, including XRD, SEM, TEM, BET, and SAXS. These mesoporous carbon samples were tested as electrode materials for supercapacitors and among the samples, OMC-meso10 was found to be the best sample with relatively high specific capacitance of 250 F/g at a current density of 1 A/g. Overall, this work has shown the porosity control of porous carbon materials by soft-templating method to enhance the supercapacitive performance for EDLCs. Therefore, this work can be accepted in this journal after addressing the following comments:

1. In the Abstract and Conclusions section, the authors need to summarize the electrochemical performance of the optimum sample for supercapacitors.

2. To understand the degree of graphitization in these ordered mesoporous carbon samples, I recommend the authors to conduct some Raman spectroscopy of analysis and possibly link that to the electrochemical performance for supercapacitors, e.g.,, if the graphitization degree is high based on the Raman analysis then the synthesized carbon has a good conductivity.

3. Stability is an important factor in supercapacitors for long-term operation of the device. Please check the stability of OMC-10.

4. In relation to Q3, the authors can check the morphological stability of OMC-1 after cycling by SEM.

5. How about the electrochemical performance of MC-0 and OMC-0 for supercapacitors? These data need to be shown as well and compared with the other electrodes (OMC-10, OMC-20 and MC-10)

6. A table comparing the specific capacitance of OMC-10 with other reported carbon electrodes can be provided.

7. In the Introduction, some recent references on the development of different electrode materials for supercapacitor applications, such as J. Chem. Technol. Biontechnol., 96, 662-671 (2021); Nano Energy, 65, 103991 (2019); Chem. Commun., 58, 1009-1012 (2022); Chem. Eng. J., 442, 136362 (2022); ACS Appl. Energy Mater., 4, 1840–1850 (2021) and Nano Convergence, 9, 10 (2022) can be introduced and cited to provide a broader perspective on the variety of electrode materials for supercapacitors.

Round 2

Reviewer 1 Report

The below issues need to be further addressed before it can be considered for publication.

1. The perspective of OMC-meso-10 in Figure 4 is not consistent with other images in Figure 4. It is strongly suggested to make them consistent for better comparison. The authors claimed that the periodic ordering of OMC-meso-10 was inferior to that of MC-meso-10. Why? It is hard to justify it from Figure 4.

2. In the captions of Figure 7 and Figure 8, the word "currant" should be revised to "current".

Reviewer 2 Report

The authors have addressed my previous comments and I am happy to accept the manuscript in the present form.

Round 3

Reviewer 1 Report

The issues raised have been addressed well and the manuscript can be accepted as it is.